# Heuristics in risky decision-making relate to preferential representation of information

Evan M. Russek [1,2,8] ✉, Rani Moran [1,2,3], Yunzhe Liu [4,5], Raymond J. Dolan [1,2] & Quentin J. M. Huys[1,2,6,7]

When making choices, individuals differ from one another, as well as from normativity, in how they weigh different types of information. One explanation for this relates to idiosyncratic preferences in what information individuals represent when evaluating choice options. Here, we test this explanation with a simple risky-decision making task, combined with magnetoencephalography (MEG). We examine the relationship between individual differences in behavioral markers of information weighting and neural representation of stimuli pertinent to incorporating that information. We find that the extent to which individuals (N = 19) behaviorally weight probability versus reward information is related to how preferentially they neurally represent stimuli most informative for making probability and reward comparisons. These results are further validated in an additional behavioral experiment (N = 88) that measures stimulus representation as the latency of perceptual detection following priming. Overall, the results suggest that differences in the information individuals consider during choice relate to their risk-taking tendencies.

When faced with a choice among actions that can lead to multiple outcomes, decision theory postulates that individuals should compute choice values by taking the expectation over the utility of outcomes, each weighted by their probability[1–3]. However, psychologists have long shown that, instead of deploying this strategy, participants exploit several heuristics, including inappropriately weighting either utility or probability information[4–7]. Although some models have offered parameterizations of heuristic reliance on either type of information[8–11], the precise neurocognitive mechanisms that underlie individual use of these heuristics remains unknown. In this work, we exploit magnetoencephelopgraphy (MEG) to test a specific hypothesis – namely, that underlying heuristic reliance on either source of information reflects a preferential representation of stimuli that are most informative for using such information during evaluation.

Recent research has pointed to selective consideration of information as a source of bias in decision making. This work has analyzed eye-tracking, mouse-tracking, and response-times to reveal that selective consideration of choice options, attributes, or other information can explain biases in value-based consumer choice[12,13], multi-attribute choice[14], social choice[15], intertemporal choice[16], and risky decision-making[17–19]. Here, we expand on this work by first providing two additional forms of evidence, based on neural decoding and behavioral priming, for a relationship between selective consideration of information and heuristic strategy in decision-making under risk. Additionally, we show an example where such selective consideration is applied to potential outcomes of a choice – thus linking this research with work on model-based simulation in planning that has looked at what outcomes of a choice individuals tend to consider when they decide[20–26].

[1]Max Planck University College London Centre for Computational Psychiatry and Ageing Research, University College London, Queen Square Institute of Neurology, London, UK. [2]Wellcome Centre for Human Neuroimaging, University College London, Queen Square Institute of Neurology, London, UK. [3]Department of Psychology, School of Biological and Behavioural Sciences, Queen Mary University of London, London, UK. [4]State Key Laboratory of Cognitive Neuroscience and Learning, IDG/McGovern Institute for Brain Research, Beijing Normal University, Beijing, China. [5]Chinese Institute for Brain Research, Beijing, China. [6]Camden and Islington NHS Foundation Trust, London, UK. [7]Division of Psychiatry, University College London, London, UK. [8]Present address: Departments of Computer Science and Psychology, Princeton University, Princeton, NJ, USA. ✉e-mail: evrussek@gmail.com

In a typical risky-choice task, individuals choose between a safe option with a known, fixed outcome, and a gamble option which can lead probabilistically to one of two possible outcomes. Normative choice in such settings requires evaluating the gamble by summing the utility of each uncertain outcome, weighted by its probability, and comparing this expected utility to the utility of a known safe option[1–3]. One explanation for deviations from normativity, as well as variability, is the need for individuals to employ heuristics that reduce the computational burden entailed in this rational approach to choice[27–31]. Whereas the normative choice strategy requires independent consideration of each possible task outcome, individuals can reduce the number of outcomes they consider through preferential reliance on a particular type of information during evaluation[9,10]. For example, individuals could prioritize probability information, and selectively ignore the safe outcome as well as the unlikely gamble outcome, leading to a decision solely based on whether the more likely gamble outcome is attractive. Alternatively, they could prioritize reward information, and solely represent outcomes useful for comparison along this dimension.

We hypothesized that prioritization of distinct types of information during choice evaluation – and more specifically preferential representation of outcome stimuli relevant for comparing choices alongside the type of information prioritized – would explain heuristic weightings of probability and reward information. We leveraged individual differences in heuristic reliance on reward or probability information in choice behavior and examined whether this variability related to inter-participant variability in a disposition to represent outcomes which support a prioritization of a one or the other type of information. If heuristic reliance on probability or reward information in behavior is related to prioritization of probability or reward information during choice evaluation, then we would expect individuals who weigh probability or reward information more in choice to preferentially represent outcome stimuli useful for comparing choices according to that information dimension. At a higher level, we sought to determine whether the outcomes that an individual tends to consider when deciding underpin the type of information their choices reflect a heuristic reliance upon.

We present affirmative evidence for this hypothesis by examining relationships between choice behavior and markers of outcome representation. Specifically, we use both MEG decoding and priming effects to determine which outcomes are preferentially represented during choice and use choice behavior to measure risk-taking heuristics. In our primary experiment, we utilize recent advances in multivariate methods for MEG[21,32–34] to decode which outcome stimuli participants represent while they make a risky choice. This involves, first, the identification of MEG signatures of visual stimuli associated with different outcomes; and, second, the examination of these signatures during choice. This data show that individual differences in outcome representation during decisions are systematically related to the individual differences in choice behavior. The secondary experiment validates this using a behavioral priming manipulation involving interruption of the choice evaluation period with a perceptual detection task (c.f.[24,35]). Consistent with our MEG experiment, this shows that faster detection of a stimulus is related to increased behavioral weighting of the information represented by that stimulus. Finally, we find that a neural marker of preferential representation is related to the real-world self-reported behavioral trait of impulsivity.

In summary, individual differences in heuristic weightings of probability and reward information during choice relate to differential tendencies related to which outcomes are prioritized for representation during option evaluation. The findings establish a link between a representation of different sources of choice relevant information and the types of decision patterns individuals manifest in risky choice.

## Results

### MEG decision-making task

Participants ($n = 19$) completed a risky decision-making task while we acquired simultaneous neural data using MEG (Fig. 1). On each trial, participants were presented with a gamble that required an accept or reject choice (Fig. 1A). Rejecting the gamble led to collection of a safe outcome, OS. Accepting led to collection of one of two gamble outcomes, O1 or O2. The chances of encountering O1 versus O2 upon acceptance of the gamble was signaled by presentation of one of four probability stimuli (P1, P2, P3, or P4; Fig. 1B). The probabilities implied by each of these stimuli were both, extensively experienced, and tested prior to task commencement (Supplementary Fig. 1). On each trial, the points paired with each outcome changed and participants were notified of the reward paired with each outcome at the start (Fig. 1C). The points paired with each stimulus was structured such that one of the two outcomes (randomly assigned on each trial to O1 or O2 in a counterbalanced manner) referred to as the trigger outcome had a reward with high absolute value, while the other gamble outcome had points close to 0. We use the term safe outcome, OS, to refer to the certain outcome whose value lay between O1 and O2. Note that for blocks involving loss trials the safe option was paired with negative points.

Critically, to facilitate MEG analysis, the time course by which information was presented was structured so as to enforce evaluation of choice options at an identifiable timepoint. Participants were first informed of the number of points paired with each outcome (Fig. 1A, left). However, this information was insufficient to make choices as the probabilities relevant to that trial were unknown at this timepoint. Choice evaluation involving the integration of outcomes O1 and O2 with their probability and comparison with the safe value could only start when the probability stimulus appeared on the screen following this (Fig. 1A, middle). Note that at this point, the outcome stimuli were no longer on the screen. Hence, we aimed to decode the neural signatures of the outcome stimuli at this time-point to determine how stimuli were represented during evaluation and how this related to the ensuing choice.

### A parameterization of heuristic reliance on probability and reward information in choice

We hypothesized that heuristic reliance on reward and probability information reflects different approaches for deciding which information to represent during evaluation. Testing this hypothesis required us to parameterize, for each participant, the extent to which choices reflected a heuristic reliance on probability versus reward information. We obtained such a parameterization using a model inspired by prior additive models fit to choices[9,10], which we refer to as the Additive Heuristic model (Fig. 2A). Applied to the current task, the Additive Heuristic model decides by computing two distinct components (Methods). A probability information component computes the relative chances that the choice stimulus will lead to the better versus worse gamble outcome. A reward information component computes the reward difference between the gamble reward midpoint and the safe reward. Note that we use the term reward to refer to number of points not only for gain trials, but also for loss trials, where a loss can be viewed as a negative reward. Importantly, because of how rewards were structured in the task (Fig. 1C), following baseline subtraction of the number of points closest to zero (so that all outcome's points are the distance from the lowest absolute point number), the difference between gamble reward midpoint and safe reward could be computed by considering just the trigger reward, which had higher absolute reward value, and the safe reward. The probability information and reward information components are respectively weighted by parameters, $\beta_{prob}$ and $\beta_{reward}$ and then added to a frame (gain or loss) specific intercept to form a choice probability (see Supplementary Fig. 2a for analysis of which parameters should be split between gain

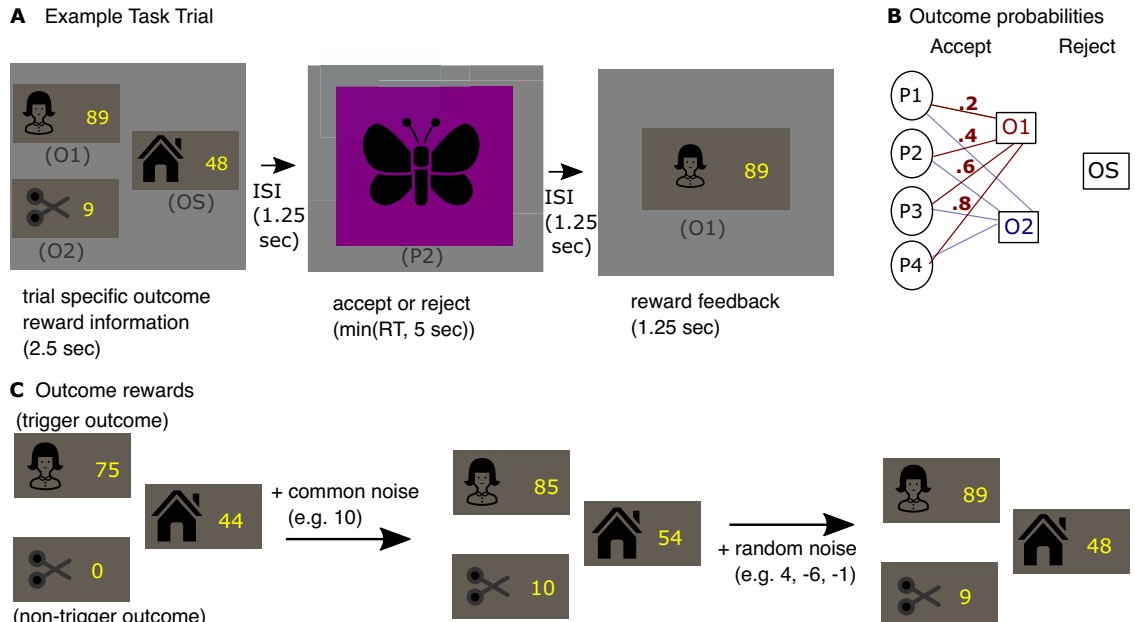

**Fig. 1 | Task. A** Example task trial. Participants chose between a safe stimulus (OS) or a gamble which probabilistically led to one of two outcome stimuli (O1 or O2). Information required for value computation was provided in discrete stages to require computation to occur at a specified time-point. Participants were first informed of the point values for all the three outcomes. Participants were then presented with one of four possible probability stimuli (P1, P2, P3 or P4) on which they had been pretrained, indicating four different probability combinations. They then decided whether to accept or reject the gamble. Rejecting led to collection of the safe outcome OS along with its trial-specific associated points. Accepting led to collecting either O1 or O2 along with the trial-specific associated points. All outcome and choice stimuli were represented by decodable visual stimuli. Note that in the example trial, the gamble was accepted. Stimuli in the real experiment were photographs. **B** Outcome probabilities. The chances of collecting O1 versus O2 upon accepting the gamble depended on which probability stimulus was presented. Probability of reaching O1 was .2, .4, .6, and .8 for P1, P2, P3 and P4 respectively, and p(O2) = 1 - p(O1). These probabilities were extensively pretrained. **C** Outcome rewards. On each trial either O1 or O2 was designated to be the 'trigger' outcome, whose value was selected from three levels (45, 65, or 75 during gain blocks or −45 −65 or −75 on loss blocks). The non-trigger outcome was always 0. OS was selected from 4 levels (20, 32, 44, 56 during gain blocks or −20, −32, −44, −56 during loss blocks). To discourage habitual responding on repeated choices, a variable amount of common noise (between 0 and 20) was added to all outcomes. Finally, a random value (between −6 and 6) was added to each outcome separately. **A, C** House and scissor images were obtained from svgrepo.com where they are published under MIT licenses. They were respectively created by Adam Whitcroft and scarlab.

and loss trials and Supplementary Fig. 2b for necessity of both reward and probability information components).

The additive heuristic model captured participants' aggregate choices in the task. It captured both deviations from a model that decided by computing expected values (Fig. 2B) and the extent to which individual participants relied on either probability or reward information in choice (Fig. 2C). The Additive Heuristic model provides two parameters for each participant, $\beta^s_{prob}$ and $\beta^s_{reward}$. These parameters measure behavioral choice reliance on probability versus reward information respectively and will henceforth be referred to as Choice Probability Weight $\beta^s_{prob(choice)}$ and and Choice Reward Weight $\beta^s_{reward(choice)}$.

Note that we do not consider this model itself provides a strong claim that valuation is additive rather than multiplicative. The choice to use the additive heuristic model is solely based on it providing a more parsimonious fit to behavior (Supplementary Fig. 3) and superior parameter identifiability (Supplementary Tables 1 and 2) compared to alternative models (see Supplementary Note 1).

### Behavioral reliance on reward versus probability information are related to distinct patterns of preferential outcome reactivation

At the group level, participants made use of both the reward and probability components of the Additive Heuristic model (Supplementary Fig. 2b). However, individuals differed substantially in their tendency to rely on one or the other component (Fig. 2C). We hypothesized that this variability reflected tendencies to consider different classes of information when evaluating choices. For example, one

means to compute the probability component of the additive heuristic model is to selectively consider the gamble outcome with higher probability, and then decide whether it was attractive. Because of the reward structure in the task (Fig. 1C) this could be determined by comparison to a fixed threshold without consideration of the other outcomes. Such a strategy could be beneficial because it could arrive at choices by forgoing consideration of both the gamble outcome with low probability, as well as the safe outcome. Conversely, the reward component could be computed by selectively considering the gamble outcome with higher absolute reward (the trigger outcome) and the safe outcome to take the reward difference between these items.

We used decoding of MEG data during choice deliberation to test whether individual variation in choice behavior was driven by differences in which outcomes individuals tended to consider. Following the above reasoning, we conjectured that individuals whose behavior reflected a greater reliance on probability information (indexed by higher Choice Probability Weight) would also tend to neurally represent gamble outcomes with higher probability. By contrast, individuals whose behavior reflected greater reliance on reward information (indexed by higher Choice Reward Weight) would tend to represent the gamble outcome with higher absolute reward and the safe outcome as this would enable them to make a comparison between the rewards of these items. Note that we will test for activation of the gamble outcome with higher probability and the gamble outcome with relative higher absolute reward in a graded manner, looking for respective effects of either probability or absolute reward on a tendency to reactivate those outcomes.

**A** Additive Heuristic Model

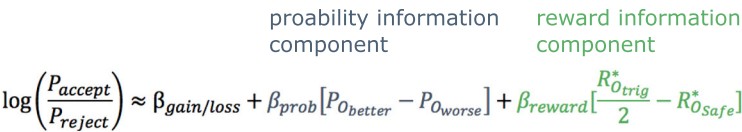

$$\log\left(\frac{P_{accept}}{P_{reject}}\right) \approx \beta_{gain/loss} + \beta_{prob}[P_{O_{better}} - P_{O_{worse}}] + \beta_{reward}\left[\frac{R^*_{O_{trig}}}{2} - R^*_{O_{Safe}}\right]$$

proability information component

reward information component

**B** Aggregate data

**C** Individual differences

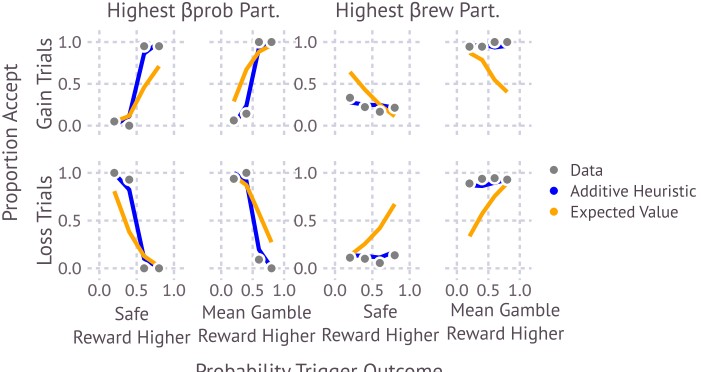

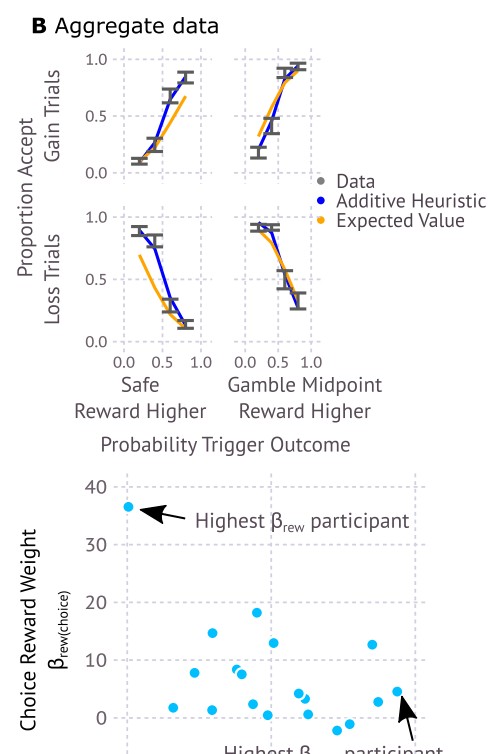

**Fig. 2 | Additive heuristic model. A** Additive Heuristic Model parameterizes use of reward and probability information. The probability information component measures the difference in probability between reaching the better (higher reward) versus worse gamble outcome, contingent on accepting the choice stimulus. The reward information component measures the difference in reward associated with the midpoint between the gamble and safe reward. Note that because of the actual reward used in the task (Fig. 1C) this difference can be computed by considering the trigger and safe rewards, without needing to refer to the non-trigger reward. R* refers to the reward after the non-trigger reward (which simply amounts to common noise along with noise specific to that outcome) has been subtracted from all rewards. Working with R*, the difference between the gamble midpoint and safe reward is computed by dividing the trigger reward by two and subtracting the safe reward. **B** Additive Heuristic Model captures aggregate patterns in choice data.

Each data error bar (gray) shows the across-participant ($n = 19$) mean (+/− s.e.m.) proportion acceptance for each combination of whether a trial is gain or loss (row), whether the safe reward is higher or lower than the midpoint between the two gamble rewards (column) and the trigger outcome probability contingent on acceptance (x-axis). Values reflect outcome rewards prior to adding common and other noise. Blue and orange lines show predictions of the additive heuristic model and expected value models, at best fit parameters. **C** Additive Heuristic Model indexes individual differences in weighting of probability and reward information. Left) Model-predictions for individual participants that either rely exclusively on probability information (left) or reward information (right). Right) Parameterization of reward and Probability Weighting ($\beta_{Prob}$ and $\beta_{rew}$) place these two participants at extreme ends of continuum over which participants ($n = 19$) vary in these two strategies.

To identify neural representations of outcome stimuli, we trained classifiers on data collected prior to the decision-making task (Fig. 3A) (see Methods). Each classifier outputted an Activation Probability, reflecting the probability that the sensor data reflected reactivation of the outcome stimulus on which it was trained (Fig. 3B; note that in the experiment stimuli were photographs). Previous research has demonstrated that different components of a stimulus representation[32], corresponding to activity at different timepoints following stimulus presentation, can reflect distinct aspects of a stimulus's representation at retrieval. On this basis we trained multiple classifiers separately on data from each 10 ms time bin, $\tau$, following the stimulus presentations. Cross-validation accuracy was quantified as the proportion of held-out trials for which the classifier corresponding to the presented outcome stimulus had the highest activation probability. We found that classifiers trained on data from $\tau = 20$ to $\tau = 500$ ms obtained above chance accuracy when tested on held out data from the same timepoint (Fig. 3C). Additionally, such classifiers were selectively accurate when tested on timepoints when they were trained (Fig. 3D). This enabled us to then investigate which aspect of an outcome's representation are reinstated during choice evaluation. Note that for further analysis, we rely on the activation probability

measure, as it is it is a more sensitive metric than discrete accuracy and can identify changes in representation even if an item is not judged to be the most likely. Finally, we additionally verified that classifiers trained to decode outcome stimuli on the localizer task maintained good accuracy at decoding outcome stimuli in the decision-making task (Supplementary Fig. 13).

We next asked which outcome representations were reinstated during choice evaluation, and related this to behavioral markers reflecting consideration of either probability or reward information. For each training timepoint from 20 to 500 ms, over which we obtained above chance classification, we applied each of the three outcome classifiers to task data from each trial from 0 to 500 ms following the presentation of the probability stimulus (Fig. 4A). This produced, for each trial, and for each outcome, a 2-d image (train timepoint $\tau$, by task/test timepoint $\tau'$), reflecting the probability that the corresponding outcome representation (at $\tau$), was reactivated at $\tau'$ following probability stimulus onset.

We first asked whether participants who relied on probability information prioritized reactivation of gamble outcomes based on their probability. We computed the difference between the (re)activation probability ($\Delta RP_O$) of O1 and O2 ($\Delta^{s,t,\tau,\tau'}_{RP_O}$ for each participant $s$,

**A** Localizer Task

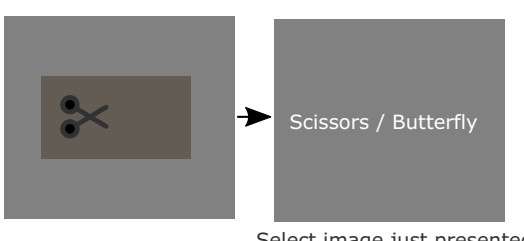

Select image just presented

**B** Classifier Output

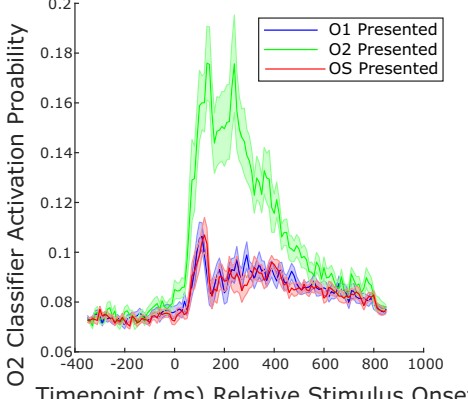

**C** Decoding Accuracy
(Train Time / Test Time Same)

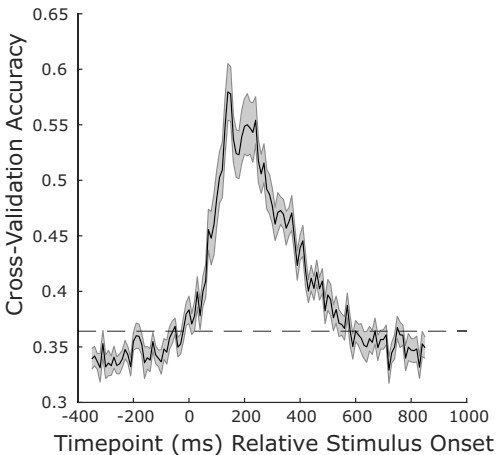

**D** Decoding Accuracy
(Temporal Specificity)

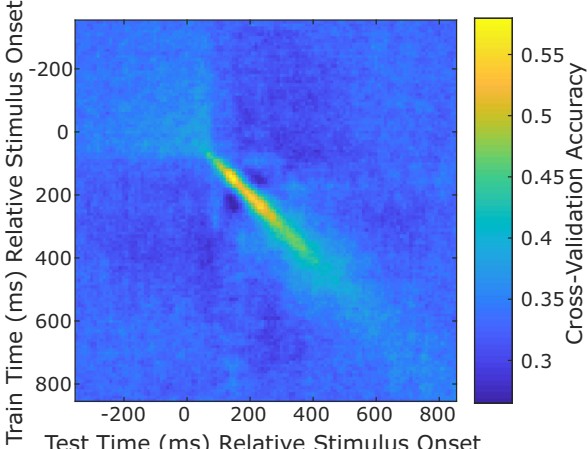

**Fig. 3 | Decoding stimulus representations from MEG. A** Localizer Task. The Localizer Task was completed prior to the risky-decision task and to learning choice-outcome probabilities. On each trial participants were shown an outcome or choice stimulus, and, on the next screen, selected a word corresponding to the stimulus they had just observed. **B** Activation Probability measure. We trained lasso-regularized logistic regression classifiers to discriminate MEG data from when a given outcome stimulus was presented compared to data from presentation of all other images and inter-trial intervals. Each classifier output an estimated probability that its stimulus was being presented (Activation Probability). Separate classifiers were trained at successive 10 ms bins of MEG data around stimulus presentation. In the example, lines display the group-mean (+/− s.e.m.) activation probability measure for the classifier corresponding to O2, for each training timepoint, applied to held out data from the same corresponding test timepoint.

Color designates the true outcome stimulus presented. **C** Decoding accuracy. Cross-validation accuracy is the proportion of trials for which the classifier corresponding to the presented outcome (for held-out data) had the highest activation probability. Lines denote mean accuracy (+/− s.e.m.) for each set of 10 ms time-binned outcome classifiers, applied to the same time-bin on held out examples. Dashed line designates permutation threshold corresponding to the 95 percentile peak threshold for accuracy lines generated with shuffled labels. **D** Temporal specificity. Classifiers trained on each 10 ms time bin were also tested on every time bin from −350 to 800 ms following presentation of stimuli from held out data. The resulting accuracy image demonstrates temporal selectivity. Classifiers identify with good accuracy representations of stimuli specific to the timepoint on which they were trained. **B**–**D** Values reflect group means across 19 participants. **A** Scissor image was created by scarlab and published on svgrepo.com under an MIT license.

trial $t$, train timepoint $\tau$, and task timepoint $\tau'$; Fig. 4B). We then fit a linear model to predict the relative reactivation measure (separately for each $s$, $\tau$, and $\tau'$) as a function of the relative probability for O1 versus O2 indicated by the choice stimulus ($\Delta_{P_O}^{s,t}$; Fig. 4C). The estimate of this effect, $\beta_{prob(neural)}^{s,\tau,\tau'}$ reflects a tendency of a participant, $s$, to prioritize reactivation of outcome representations (elicited $\tau$ following their direct presentation) according to their probability (measured at $\tau'$ following probability stimulus presentation; Fig. 4D). We refer to $\beta_{prob(neural)}^{s,\tau,\tau'}$ as Neural Probability Prioritization.

To test whether a tendency to reactivate outcomes according to their probability is reflected in a behavioral choice weighting of outcome probability information, we computed the between-participant relationship between Neural Probability Prioritization, $\beta_{prob(neural)}^{s,\tau,\tau'}$ and Choice Probability Weight, $\beta_{prob(choice)}^{s}$ (Fig. 4E). The peak of this effect was significantly positive (Fig. 4F, G $\tau = 420$ ms; $\tau' = 420$ ms;

$t_{peak} = 3.024$, $P_{FWE} = .010$, one-tailed non-parametric permutation test on image peak; see Methods; see Discussion for consideration of identified peak significant timepoints; see Supplementary Note 2, Supplementary Figs. 8, 9a for estimation of unbiased behavioral-neural correlations), supporting the hypothesis that the more an individual's reactivation reflected differences in outcome probabilities, the more that individual showed behavioral evidence of sensitivity to probability information. Importantly, the relationship between $\beta_{prob(neural)}^{s,\tau,\tau'}$ and $\beta_{rew(choice)}^{s}$ was not statistically significant (Supplementary Fig. 7a).

In a similar manner, we investigated the reward component, which calls for consideration of the trigger outcome (gamble outcome with higher absolute reward) and safe outcome value (Fig. 2A). Thus, we asked whether individuals who were more behaviorally sensitive to reward information preferentially reinstated these outcomes. To measure a tendency to reactivate gamble outcomes with higher

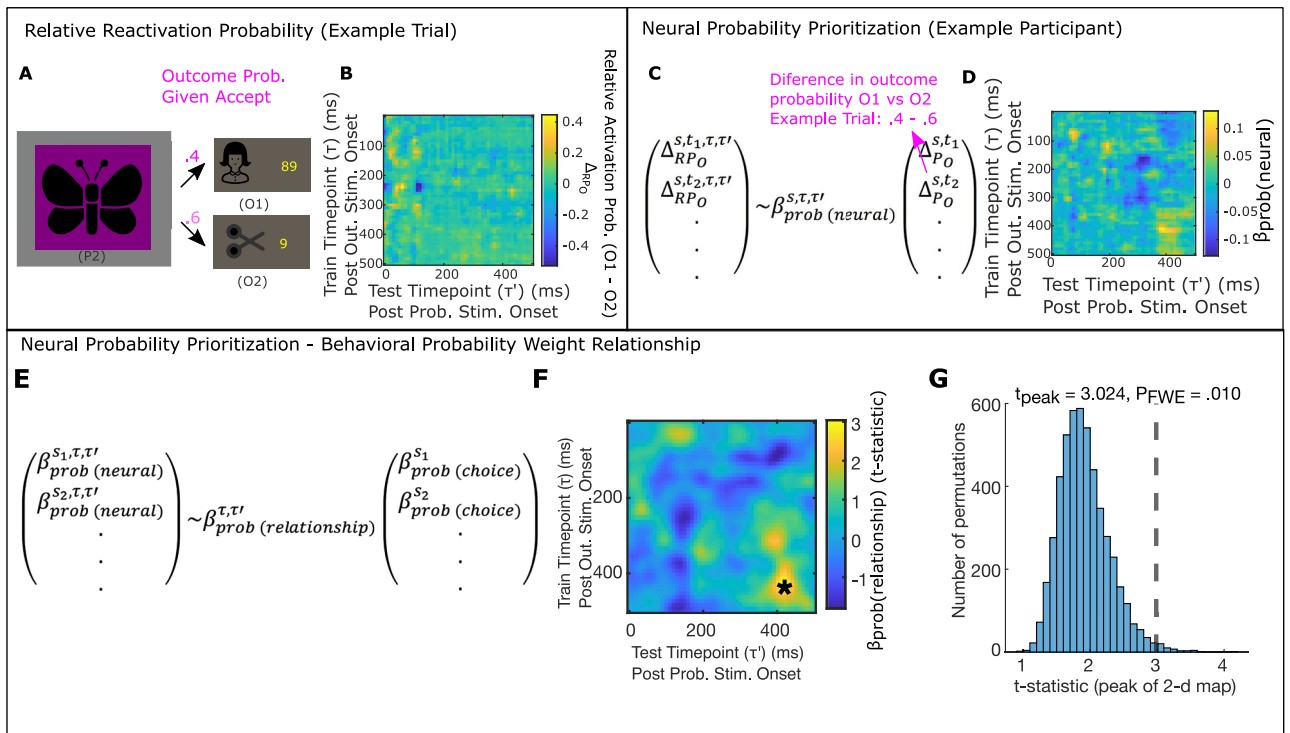

**Fig. 4 | Behavioral weighting of probability information relates to relative activation of more probable gamble outcome representation. A–D** Neural Probability Prioritization, $\beta_{prob(neural)}^{s,\tau,\tau'}$, measures dependence of activation probability on relative outcome probability. **A** In this example trial, (Trial 2 from Participant 11), P2 was presented, indicating that, if accepted, O1 would be reached with .4 probability and O2 would be reached with probability 0.6. **B** Following probability stimulus presentation, we measure relative activation probability for O1 and O2, $\Delta_{RP_O}^{s,\tau,\tau'}$, for $\tau' = 0$ to $\tau' = 500$ ms following probability stimulus onset, classifiers trained on MEG sensor data from $\tau = 20$ to $\tau = 500$ ms following outcome stimulus onset in the localizer task. Image demonstrates the results of this computation for the example trial in (**A**). **C** Neural Probability Prioritization, $\beta_{prob(neural)}^{s,\tau,\tau'}$, is computed by regressing relative trial-varying activation probability of O1 versus O2, $\Delta_{RP_O}^{s,t,\tau,\tau'}$, onto the trial-varying probability of encountering O1 versus O2, $\Delta_{P_O}^{s,t}$ (see Methods).

**D** Image denotes $\beta_{prob(neural)}^{s,\tau,\tau'}$ for every classifier train timepoint,$\tau$, following outcome stimulus onset and test timepoint,$\tau'$, following probability stimulus onset, for an example participant (s = 11). **E–G** Choice Probability Weight relates to Neural Probability Prioritization. **E** We measured the between-participant relationship between $\beta_{prob(neural)}^{s,\tau,\tau'}$ and behavioral evidence for consideration of probability information, $\beta_{prob(choice)}^{s}$ by regressing $\beta_{prob(neural)}^{s,\tau,\tau'}$ onto $\beta_{prob(choice)}^{s}$, separately for each train and test timepoint, $\tau$ and,$\tau'$. **F** T-statistic for this regression (applied to 19 participants), for each train and test timepoint, smoothed with a Gaussian kernel ($\sigma = 1.5$ timebins). *$P_{FWE} = .010$, one-sided non-parametric permutation test on image peak. **G** Histogram shows null distribution of maximum t-statistics over 5000 2-d maps, each generated by randomly shuffling $\beta_{prob(choice)}^{s}$ between participants, $s$. Dashed line shows measured maximum t-statistic. **A** Scissor image was created by scarlab and published on svgrepo.com under an MIT license.

absolute reward values, we measured the between-trial effect of the difference between the absolute rewards for O1 and O2, $\Delta_{|R_O|}^{s,t}$, on the difference in (re)activation probability for O1 and O2, $\Delta_{RP_O}^{s,t,\tau,\tau'}$ (Fig. 5A–C). This effect, $\beta_{rew(neural)}^{s,\tau,\tau'}$, Neural Reward Prioritization, measures a participant's tendency to prioritize reactivation of an outcome's representation (at specific $\tau$ and $\tau'$) based on its trial-varying absolute reward value (Fig. 5D). Regressing $\beta_{rew(neural)}^{s,\tau,\tau'}$ onto Choice Reward Weight ($\beta_{rew(behavior)}^{s}$; Fig. 5E), revealed a significant positive effect (Fig. 5F–G $\tau = 480$ ms, $\tau' = 110$ ms; $t_{peak} = 2.974$, $P_{FWE} = .034$, one-tailed non-parametric permutation test on image peak; see Supplementary Fig. 9b for estimation of unbiased behavior-neural correlation). This association was specific as the relationship between $\beta_{rew(neural)}^{s,\tau,\tau'}$ and $\beta_{prob(choice)}^{s}$ (Supplementary Fig. 7b) was not statistically significant.

We additionally computed participant specific tendencies to reactivate the safe outcome OS, $RP_{O_S}^{s,\tau,\tau'}$, as the mean reactivation probability of the safe outcome classifier across trials (Supplementary Fig. 6a) and regressed this onto Choice Reward Weight ($\beta_{rew(choice)}^{s}$; Supplementary Fig. 6b). Although the peak of this effect was also significantly positive ($\tau = 350$ ms $\tau' = 270$ ms; $t_{peak} = 5.464$, $P_{FWE} = .011$, one-tailed non-parametric permutation test on image peak; Supplementary Fig. 6c, d), we found that this effect was dependent on a single participant which warrants caution in interpretation (Supplementary Fig. 9c). As with the above, we did not observe a statistically significant

positive relationship between $RP_{O_s}^{s,\tau,\tau'}$ and $\beta_{prob(choice)}^{s}$ (Supplementary Fig. 7c).

These effects suggest that the more an individual relied on a simple comparison between the rewards from a gamble and safe options, the more they reactivated the high absolute reward outcome. Additionally, we find weak evidence that these individuals also activated the safe outcome, as would be expected for a comparison.

We additionally employed an approach previously used in Ref. 32 to determine the sensors responsible for driving reactivation events responsible for both relationships between Neural Probability Prioritization and Choice Probability Weight, and between Neural Reward Reactivation and Choice Reward Weight. Specifically, we repeated each of these analyses 5000 times, each time using a subset of 50 randomly drawn sensors, and then performed a regression to determine weights measuring each sensor's contribution to the observed effect. This revealed reactivation events underlying both effects to rely primarily visual and temporal sensors, with a small number of frontal sensors contributing as well (Supplementary Fig. 10).

All effects found were specific to locking on the time of the Probability Stimulus presentation and were not statistically significant when aligning events to the time of response (Supplementary Fig. 11). Although our analysis here relies on relating between participant variability between neural and behavioral markers of prioritization, we also observe trending evidence for main effects of Neural Probability

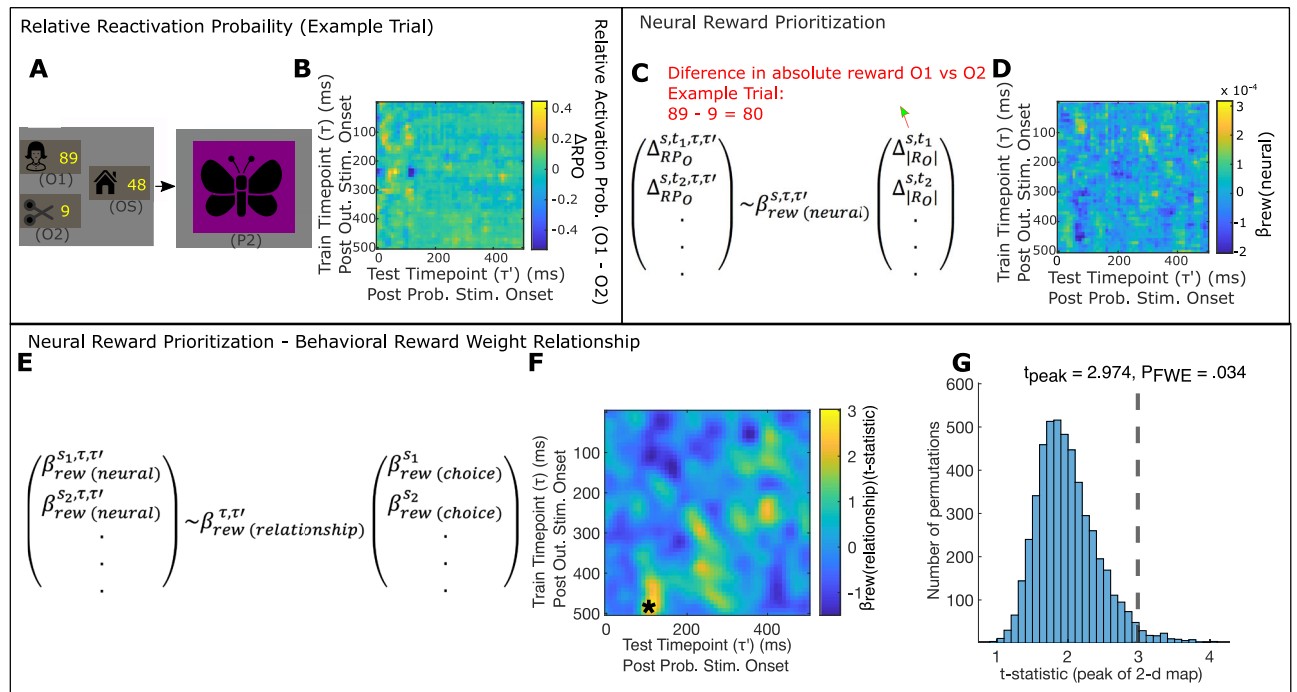

**Fig. 5 | Behavioral weighting of reward information relates to relative activation of higher absolute reward gamble outcome representation. A–D** Neural Reward Prioritization, $\beta_{rew(neural)}^{s,\tau,\tau'}$, measures dependence of reactivation probability on relative outcome absolute reward. **A** In this trial, O1 is paired with 89 points and O2 is paired with 9 points. Note that this is the same trial as in Fig. 4A. **B** Image displays $\Delta_{RP_O}^{s,t,\tau,\tau'}$ for example trial in 5a. Replotted from Fig. 4B. **C** Neural Reward Prioritization, $\beta_{rew(neural)}^{s,\tau,\tau'}$, is computed by regressing relative trial-varying reactivation probability of O1 versus O2, $\Delta_{RP_O}^{s,t,\tau,\tau'}$, onto the trial-varying difference in absolute points paired with O1 versus O2, $\Delta_{|R_O|}^{s,t}$. **D** Image denotes $\beta_{rew(neural)}^{s,\tau,\tau'}$ for every classifier train timepoint, τ, following outcome stimulus onset, and test timepoint, τ′, following probability stimulus onset, for an example participant (s = 11). **E–G** Choice Reward Weight relates to Neural Reward Prioritization. Following computation $\beta_{rew(neural)}^{s,\tau,\tau'}$, we measured the between-participant relationship between this and behavioral sensitivity to reward information, as measured by Choice Reward Weight, $\beta_{rew(choice)}^{s}$. This was done by regressing $\beta_{rew(neural)}^{p,\tau,\tau'}$ onto $\beta_{rew(choice)}^{s}$ separately for each τ and τ′. **F** Image shows a t-statistic for this regression (across 19 participants), for each train and task time-bin, smoothed with a Gaussian kernel (σ = 1.5 time-bins). *: $P_{FWE}$ = .034, one-sided non-parametric permutation test on image peak. **G** Histogram shows null distribution of maximum t-statistics over 5000 2-d maps, each generated by randomly shuffling $\beta_{rew(choice)}^{s}$ between participants. Dashed line shows true maximum t-statistic. **A** House and scissor images were obtained from svgrepo.com where they are published under MIT licenses. They were respectively created by Adam Whitcroft and scarlab.

Prioritization and Neural Reward Prioritization in participants with high Choice Probability Weights and Choice Reward Weights respectively (Supplementary Fig. 12). Finally, these behavioral-neural effects were robust to different choices for data preprocessing and to the addition of covariates controlling for between session changes in decoding accuracy (Supplementary Figs. 15 and 17).

Altogether, these results support the idea that individual differences in outcome reactivation prioritization relate to individual differences in choices. Participants who were behaviorally reliant on probability information were also more likely to reactivate gamble outcomes based on their probability. Conversely, participants who were behaviorally reliant on reward information tended to reactivate gamble outcomes based on their absolute reward. Hence, whether probability and reward information influenced behavior related to preferential neural reactivation for the relevant dimension of information.

## Perceptual detection task

We next sought to conceptually replicate these findings using a robust behavioral measure. Inspired by work on priming, as well as work which has used response times to identify prospective representations of stimuli[24,35], we reasoned that the active representation of stimuli (as identified in the MEG study) should influence the speed at which those stimuli are perceptually detected. Based on this reasoning we devised a new version of the task where we used a priming and perceptual detection manipulation to index representation.

The task was equivalent to the decision task used in the MEG study, except for the use of perceptual detection rather than MEG decoding to index which outcomes were represented during choice. The key departure was that on one-third of trials participants performed a perceptual detection task rather than a choice task (Fig. 6A). Specifically, following presentation of the probability stimulus, participants were presented with a screen showing the three outcome stimuli. One of the stimuli contained a probe – an arrowhead symbol – and participants were required to report, as quickly as possible, the direction of the arrow.

Similar to MEG decoding, the detection of the arrowhead direction amongst the stimuli offers an opportunity to ascertain the extent to which the probed stimulus was being actively represented during presentation of the Probability stimulus. If participants were actively representing a stimulus, then processing of that stimulus would be prioritized and this would result in faster detection of the probe direction when the probe was placed on that stimulus. Using this approach, we tested whether a tendency to use probability versus reward information in choice was accompanied by a tendency to prioritize outcomes for reactivation based on either their probability or absolute reward, as indicated by response times.

Analogous to our MEG analysis (Figs. 4 and 5) we measured the extent to which the probed stimulus's probability and absolute reward affected response times in reporting the probe ($\beta_{prob(RT)}$ and $\beta_{rew(RT)}$ respectively). More negative values of $\beta_{prob(RT)}$ and $\beta_{rew(RT)}$ indicate that participants were more inclined to represent outcome stimuli when they had higher probability, or higher absolute reward

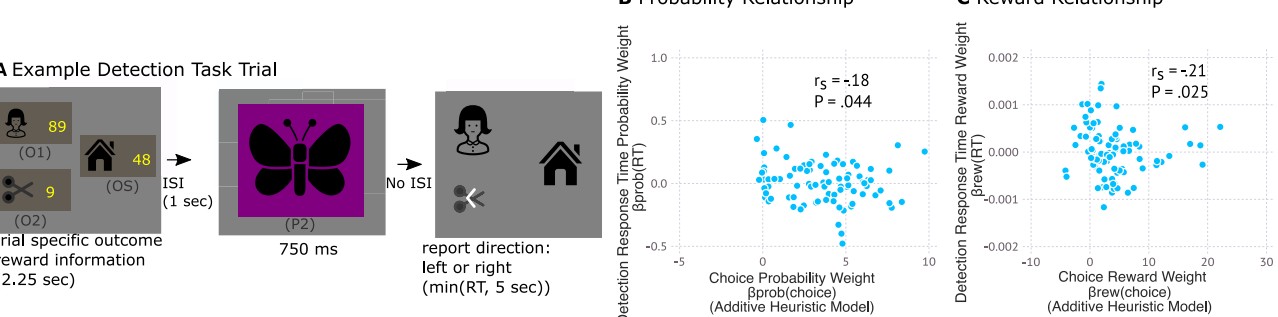

**Fig. 6 | Perceptual detection task provides conceptual replication of key findings. A** Example task trial. In the perceptual detection task, two thirds of trials were equivalent to trials in the MEG decision making task (Fig. 1A). In one third of trials, following presentation of the probability stimulus (750 ms), the probability stimulus was removed and the three outcome stimuli were presented. One of the these (the probed stimulus) had an arrow placed upon it and the participant was required to respond as quickly as possible to report the arrow direction. **B** Choice Probability Weight relates to response time marker of probability prioritization. Response Time Probability Weight, $β_{prob(RT)}$, measures the effect of the probe stimulus's probability (conditioned on accepting the gamble presented earlier in the trial) on the participant's response time in reporting the arrow's direction. Negative values for $β_{prob(RT)}$ indicate faster responses when the probed stimulus is more probable, indicative of the more probable outcome being represented during the Probability stimulus presentation. We observed a negative relationship between $β_{prob(RT)}$ and $β_{prob(choice)}$ which measures weighting of probability information in choice. This provides a conceptual replication of the MEG findings in Fig. 4. **C** Choice Reward Weight relates to response time marker of reward prioritization. Response Time Reward Weight, $β_{rew(RT)}$, measures the effect of the probe stimulus's absolute reward (relative to the other gamble outcome) on the participant's response time to report the arrow probe. We observed a negative relationship between $β_{rew(RT)}$ and $β_{rew(choice)}$ which measures weighting of reward information in choice. This provides a conceptual replication of MEG findings from Fig. 5. **B, C** P-values reflect one-sided t-test for spearman correlation. Tests reflect planned comparisons and thus are not corrected. **A** House and scissor images were obtained from svgrepo.com where they are published under MIT licenses. They were respectively created by Adam Whitcroft and scarlab.

respectively. We then tested whether these tendencies were related to use of probability versus reward information in forming decision variables.

We found that a greater tendency to use probability information in choice (measured as utilizing a greater Choice Probability Weight, $β_{prob(choice)}$) related to a greater tendency to represent outcomes based on their probability (measured as lower $β_{prob(RT)}$ reflecting faster responses for more probable outcome stimuli; spearman rank correlation, one-tailed; $r_{spearman} = $ -.183, $t(86) = −1.72$, $P = .044$; Fig. 6B). This provides a conceptual replication of the MEG results presented in Fig. 4.

Analogously for reward, a greater tendency to use reward information in choice (measured as utilizing a greater Choice Reward Weight, $β_{rew(choice)}$) related to a greater tendency to represent outcomes based on their reward (measured as lower $β_{rew(RT)}$ reflecting faster responses for outcome stimuli with higher absolute reward; Fig. 6C; spearman rank correlation, one-tailed, $r_{spearman} = $ -.21, $t(86) = −2.0$, $P = .025$). This provides a conceptual replication of the MEG results presented in Fig. 5.

### Preferential reactivation of high probability outcomes relates to a real-life measure of risky decisions

Aberrant valuation and decision making, particularly in risk settings, are features of multiple psychiatric disorders[36–41]. Based upon the finding above, we hypothesized that aberrant decision making and valuation in the context of behavioral impulsivity tendencies would relate to a lack of selectivity in reactivation of choice outcomes. Impulsivity is characterized by a predisposition toward risky behavior and a predisposition to act without adequate thought[42]. Items on the self-report Barratt Impulsivity Scale (BIS) capture a tendency to act without thinking about the likely future consequences of the action (e.g. I do things without thinking, I am more interested in the present than the future). Impulsivity has also previously been associated with reduced neural signatures of model-based decision making[38], while theoretical models of impulsivity suggest a relationship between it and noisy simulation of action outcomes[43]. Based on this, we specifically hypothesized that impulsivity would relate to failure to reactivate (consider) outcomes according to their probability. We thus examined the relationship between impulsivity and Neural Probability Prioritization ($β_{prob(neural)}^{s,τ,τ'}$, Fig. 7A) and identified a significant negative relationship (Fig. 7B, C, $τ = 410$ ms, $τ' = 370$ ms, $P_{FWE} = .001$, one-tailed non-parametric permutation test on image minimum; see Supplementary Fig. 9d for estimation of unbiased behavior-neural correlation).

In relation to this result we caution that because probability and reward information are always presented in the same order, we cannot entirely rule out that a reduced representation of high probability outcomes in individuals with higher impulsivity might in fact reflect it being presented as the second piece of information, rather than the first. Additionally, we did not identify a similar significant relationship in the perceptual detection task. Specifically, the relationship between (slower) response times for perceptual detection for higher probability probe items and participant self-reported BIS score was not significant (spearman rank correlation; one-tailed, $r_{spearman} = $ -.084, $t(86) = − .78$, $P = .79$).

## Discussion

It is widely conjectured that differences in behavioral choice patterns relate to differences in what information individuals consider during evaluation. Here, we examined this question behaviorally and with neural data. Our findings are consistent with a hypothesis that individual differences in integration of reward and probability information into choice, in both a laboratory task and in real life, reflect differences in the nature of the information that is prioritized during evaluation.

Our behavioral analysis revealed that participants differed in the extent to which they relied on either reward versus probability comparisons when deciding. By decoding outcome representations using MEG, we show these distinct decision strategies reflected differences in what outcomes were neurally represented during evaluation. Participants who decided based on a difference in probability between the better and worse gamble outcomes preferentially reactivated high probability gamble outcomes, suggesting they primarily considered probability information. Conversely, participants who decided more based on the difference in reward between outcomes preferentially reinstated the high absolute reward gamble outcomes, suggesting

### Neural Probability Prioritization - Behavioral Impulsivity Relationship

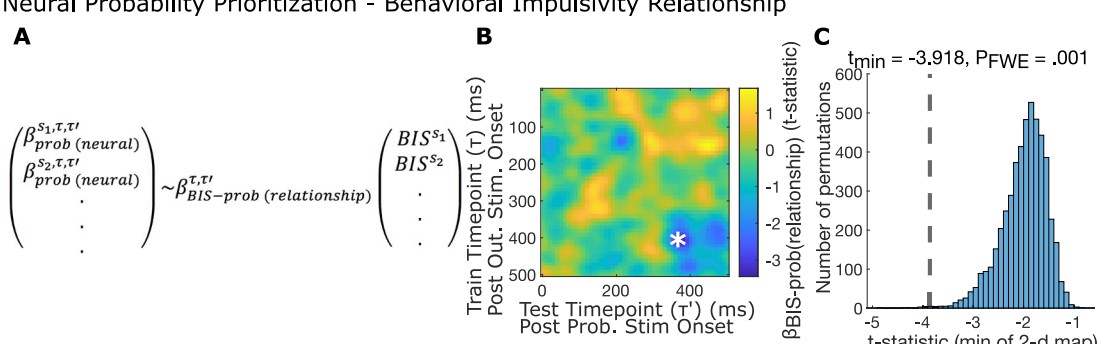

**Fig. 7 | Higher behavioral impulsivity is associated with less relative activation of high probability outcomes. A** Measuring the relationship between Behavioral Impulsivity and Neural Probability Prioritization. To measure the between-participant relationship between behavioral impulsivity and neural probability prioritization, we regressed between participant neural probability prioritization $\beta_{prob(neural)}^{p,\tau,\tau'}$ onto Behavioral Impulsivity Scale (BIS) scores, separately for each train and test timepoint ($\tau$ and $\tau'$). **B, C** Behavioral Impulsivity relates to Neural Probability Prioritization. **B** Image of t-statistic of relationship (for 18 participants) between Behavioral Impulsivity Scale (BIS) score and neural probability prioritization, $\beta_{prob(neural)}$, computed for each train and test timepoint, smoothed with a Gaussian kernel ($\sigma = 1.5$ time-bins). *: $P_{FWE} = .001$, one-sided non-parametric permutation test on image minimum. **C** Histogram shows null distribution of minimum t-statistics over 5000 2-d maps, each generated by randomly shuffling BIS scores between participants. Dashed line shows true minimum t-statistic.

they mainly considered the relative reward of gamble options with the safe option.

Our results add to the literature as to what accounts for individual differences in the treatment of reward and probability during risky choice. Although individual differences are ubiquitous in the literature of risky choice, the full range of factors that determine individual differences are unknown. Previous modeling approaches have demonstrated that models which preferentially integrate either reward or probability information account for some aspects of commonly observed variance in risky choice[9], though whether such variation is explained by differences in the types of outcomes considered during choice evaluation has not been shown. Here, by identifying a link between outcomes that are represented during choice evaluation and behavioral signatures that reflect consideration of either reward or probability information, we provide evidence that this variation is related to the types of information prioritized during evaluation. These results add to recent research demonstrating that heuristics and biases in choice can be explained for by individual differences in the types of information considered during the decision process[12–18] and demonstrate a link between these processes and previous research demonstrating the role of outcome reactivation in choice evaluation[21,22,24–26,44].

One caveat to our reactivation results is that we only analyzed choice periods of up to 500 milliseconds following choice stimulus presentation. This was necessary because participants made fast responses (Supplementary Fig. 5), limiting the available time window over which activations could be averaged. However, most participant's choice evaluations lasted longer than this time period, suggesting that we only examined reactivation data corresponding to a fraction of the possible evaluation time used by participants. One explanation for an apparent success in identifying relationships between reactivation and behavior, despite not including the entire evaluation period, is that outcome consideration at a neural level unfolded immediately upon choice stimulus onset, possibly at stereotyped time-points, and then continued beyond that until a choice was made. Notably, the same behavioral-neural relationships were not statistically significant when locking events to response times (Supplementary Fig. 11). Although we were limited to examination of the fastest reactivation measures that cohered across participants, future studies might avail of other methods. For example, identification of transitions in reactivation events between stimuli[33] may enable aggregation of reactivation events across trials that may have different response times, thus availing of all evaluation data.

Although our results support a hypothesis that heuristic reliance on probability versus reward information is driven by which outcomes are represented during choice, a major caveat is that our evidence is correlational and does not support a causal conclusion. Future work could assess the latter by causally manipulating which outcomes are represented during choice, perhaps by priming participants to attend to one or other outcome by including additional outcome features.

An additional aspect of our design is that probability versus reward information was always presented in the same order. This does not impact interpretation of our results because we did not seek to determine whether probability versus reward information is represented to a greater degree in general. Instead, our goal was to ascertain whether individual differences in representation of such information relates to individual differences in use of either source of information at choice. One potential exception to this is our finding that self-reported impulsivity relates to a lesser representation of outcomes based on their probability. We acknowledge that a reduced representation of high probability outcomes in individuals with greater BIS scores could be explained by it being presented as the second piece of information, rather than the first, if individuals with greater BIS scores preferentially represented earlier compared to later information.

A key aspect of our design was its inclusion of only two gamble outcomes. This was motivated by two considerations. Firstly, we wanted to render our task directly comparable to prior work which has characterized choice biases using two outcomes[8–11]. Secondly, we wanted to enable the simplest possible decoding analysis of outcome representation reactivations, such that two gamble outcomes can be compared. Although including two outcomes was beneficial for the decoding analysis, future work might utilize tasks with additional outcomes to enable a more fine-grained examination of how outcomes are prioritized for representation, and how this relates to choice heuristics.

A final limitation of the MEG study is that due to interference of the Covid-19 pandemic the sample size ($N = 19$) is relatively low. Although this number of participants is less than intended at study inception, we note that it is close to the range for similar studies in the field[22,45,46]. Recent work examining power in between-participant neuroimaging analysis has shown that amount of per-participant data contributes equally to power as number of participants[47]. In this regard, the amount of per-participant data in this study is relatively high, with 288 decision trials per participant. Finally, we note that the additional behavioral perceptual detection study provides a conceptual replication of the MEG results. Regardless however, the

relatively low number of participants warrants some caution in interpretation of the MEG results. Further work on this topic with higher between-participant power would be valuable.

Several previous studies have investigated outcome reactivation in the context of model-based reinforcement learning algorithms[22,24–26,44]. Typical model-based algorithms postulate that choices are evaluated by simulating potential consequential outcomes and by adding rewards from these outcomes to a running average[48,49]. Evidence that outcome reactivation functions to simulate outcomes in this manner comes from studies demonstrating that variation in a tendency to reactivate the deterministic outcome of a chosen action predicts a propensity for behavior to reflect model-based choice evaluation[21,22]. This mechanism for outcome reactivation also accounts for within participant variation of what is simulated to ultimate valuation of a choice option[26,44]. Our results add to this work by revealing that outcome reactivation can support functions beyond typical model-based simulation, such as comparison of reward values between choice outcomes (as used in the reward component of the choice model identified here). Furthermore, our results show that individual variation in reactivation tendencies relate to individual differences in choice. The results point toward a more general flexibility in the computational function of outcome reactivation and emphasize a close link between the processes determining reactivation and ultimate behavior.

Relatedly, a recent body of work has examined how the brain solves the meta-decision problem as to which potential outcomes of a choice should be simulated. Although standard formulations of simulation in model-based choice postulate that outcomes should be simulated proportionally to their probability[48] theoretical analyses have demonstrated that in situations where the total number of simulations is limited, it is possible to arrive at more accurate estimates of choice utility by a consideration of outcome utilities in the decision of what to simulate[50,51]. With MEG, tendencies to reactivate outcomes either proportionally to their utility or inverse utility have also been reported[26,52].

Our use of MEG rather than fMRI permitted an analysis of not only which outcomes were reactivated, but also the temporal structure of when such reactivations occur and what temporal component of a representation, in terms of time following direct presentation of the stimulus, was reactivated. Such temporal structure has previously been demonstrated as important for integration of rewards with non-directly paired stimuli in a sensory pre-conditioning task[32]. Our findings as to when reactivation events occur bears similarities to that previous study. Notably, our identification of reactivation related to integration of reward and probability information, occurred at two distinct timepoints (110 ms and 420 ms following choice stimulus onset), approximately resembling time-points[32] when a non-direct rewarded stimulus was re-activated following a paired stimulus onset (400 ms) or a reward (70 ms). Relatedly, we found that activated representations were those corresponding to classifiers trained around 400 ms following stimulus onset. Previous work[32] has identified such classification time-points corresponded to representations that load on temporal cortex topographies, suggesting these areas may support decision-relevant outcome representations. In our analysis of sensors responsible for reactivation events underlying behavioral-neural relationships, we also found that reactivation involved visual and temporal sensors, with some frontal sensors involved as well (Supplementary Fig. 10).

Finally, we identified that participants with higher behavioral impulsivity demonstrated relatively reduced prioritized reactivation of higher probability outcome representations. This reactivation result matches recent theoretical proposals that impulsive choice may result from a noisy simulation of future events[43]. Given the separate, positive relationship of this pattern of reactivation with integration of probability information, this points toward a potential mechanism to explain real-life aberrant risky choice, potentially a neglect of probability information[53]. More generally, our finding here opens line of research that disorders of choice may relate to what information should be prioritized. However, we note that our failure to replicate this significant result in a perceptual detection study suggests that it needs to be further investigated.

In summary, we demonstrate a relationship between the nature of the information individuals tend to consider during evaluation, and how they decide. This implies that one could learn to make better choices by learning to change what information is prioritized for consideration and points toward a research direction for treatment of mental health disorders characterized by aberrant choice.

## Methods
### MEG study experimental procedures
This study was approved by the UCL ethics board (ID: 9929/002).

**Participants.** We recruited 21 participants (mean (std) age: 23.67 (4.33), 13 female) from University College London participant databases. Participants provided informed consent prior to beginning the study. 13 were female. The mean age was 23.67, with a range of 18 to 36. No statistical method was used to predetermine sample size. Based on consideration from prior literature, we chose a sample of 30 participants, however, due to the coronavirus pandemic and the UK lockdown, we were required to stop collecting data at 21 participants. Two participants were removed from analysis for choosing the same action on greater than 80% of trials (89% and 83%), thus leaving 19 participants included in the main analysis (Figs. 2–5). We additionally failed to collect questionnaire data for one participant. Thus the neural-questionnaire analysis (Fig. 7) reflects data from 18 participants. For completing the entire study, participants were paid 40 GBP with a performance dependent bonus of up to 20 GBP.

**Task overview.** The entire task took place over two consecutive days. On day 1, participants completed the task instructions. Following this, using different stimuli than used in the actual task, participants completed the entire probability learning task, and then completed three randomly selected blocks from the risky decision-making task. Following this they completed a number of Questionnaires. Note that participants completed day 1 from their own personal computers and that behavior from practice trials on day 1 was not analyzed and is not reported here.

On day 2, in the MEG scanner, participants completed the functional localizer task, the probability learning task, and the risky decision-making task. Different task stimuli were used on Day 1 and Day 2. The full MEG session lasted about 90 minutes and consisted of 13 runs of scanning sessions. This included 3 runs of the localizer task (each lasting about 5 minutes), 2 runs of probability learning task (each less than 5 minutes, not analyzed), and 8 runs of the decision-making task (each lasting about 7 minutes).

In the main task, participants were required to make decisions about whether to accept or reject a gamble (Fig. 1A). Rejecting the gamble led to collecting a safe outcome (OS). Accepting, in contrast, led to collecting one of two gamble outcomes (O1 or O2). On each trial, each of the three outcomes were associated with a distinct number of points, which the participant was made aware of at the start of the trial, and which, if collected, contributed toward a bonus. The task contained four probability stimuli (P1, P2, P3 and P4). Each probability stimulus determined, whether, if accepting the gamble, the probability that O1 versus O2 would be encountered. The probability of gamble acceptance leading to O1 was .2, .4, .6, and .8, for P1, P2, P3 and P4 respectively (Fig. 1B).

Note that our decision to include two potential gamble outcomes in the task was based on two reasons. The first was to make our task directly comparable to prior work which has characterized choice

biases in tasks using two outcomes[8–11]. The second was to enable the simplest possible decoding analysis of representation reactivations, such that activations of two gamble outcomes could simply be compared. Although this decision to include two outcomes was beneficial toward making decoding analysis simpler, future work might utilize tasks with additional outcomes so as to study in a more fine-grained manner how outcomes are prioritized for representation and how this relates to choice heuristics.

The task consisted of eight blocks, which alternated between gain and loss blocks (four of each). To construct each trial, either O1 or O2 was selected to be the trigger option. The reward value of the trigger option was selected from {47.5, 60, 75} on gain trials, or {-47.5, -60, -75} on loss trials, and the non-trigger option value was 0 (Fig. 1C). The value of the safe option was selected from {20, 40, 60, 80} on gain trials and {-20, -40, -60, -80} on loss trials. Following this, a single random value drawn from uniform(0,20) for gain trials, or uniform(-20,0) for loss trials was added to each outcome. Finally, three separate random values drawn from uniform(0,5) for gain trials and uniform(0,-5) for loss trials were added to each value separately.

Trials consisted of each combination of trigger value, and safe value, such that the absolute value of the trigger value was greater than the absolute value of the safe value, for both O1 and O2 occurring as the trigger value, for each level of P(O1|Cn). Finally, each exact trial repeated twice in the task.

Participants were instructed that their bonus would be computed by randomly selecting one trial from each block of the task and adding the points they collected on these trials. The bonus was proportional to this sum.

**Functional localizer task.** For the functional localizer, each task stimulus was represented using a decodable visual stimulus. These consisted of photographs whose categories are pictorially represented in Fig. 1A. Our analysis of the task relied on decoding from MEG data what outcome stimulus was represented during choice evaluation. In order to collect data with which to train a classifier to detect stimulus representations, participants completed a functional localizer task, consisting of three blocks. Each block, the seven images representing each task state were each presented 20 times, in randomized order. For each presentation, the image was presented for 800 ms. Following a 200 ms ISI, two words appeared on the screen, one corresponding to the name of the image just presented and one corresponding to the name of a different image. Participants were given 600 ms to select the word corresponding to the image just seen.

**Probability learning task.** To learn the probabilities that each choice stimulus, if accepted, led to either gamble outcome stimulus, participants completed four blocks of a probability learning task. In each block, for each probability stimulus, participants were first shown a screen instructing them on the probabilities that that probability stimulus (if as part of a gamble that was accepted) would lead to either gamble outcome stimulus. Following this, the participants experienced 10 trials in which they were required to play that probability stimulus. For each play, the participant experienced that stimulus, followed by one of the two gamble outcomes. For the 10 trials, it was guaranteed that the number of either outcome experienced matched the instructed probability, however in randomized order (e.g. if the probability stimulus led to O1, 40% of the time, the participant experienced O1 4 out of the 10 times following the choice stimulus). In order to ensure attention, following 25% of these trials, participants were required to report either which choice stimulus, or which outcome stimulus they had just experienced. After experiencing two rounds of instructed probabilities and experienced transitions for each probability stimulus, the participant was then required to respond to a number of queries about the probability that each probability stimulus led to each outcome. For each query, the participant was shown an image of one of outcome stimuli as well as two of the probability stimuli, and was required to report which of the two probability stimuli was more likely to lead to that outcome. The proportion correct for these queries across rounds is reported in Supplementary Fig. 1.

**Risky decision-making task.** For the main, risky decision-making task, on each trial participants were first shown how many points would be earned if they were to encounter either of the three types of outcomes (O1, O2 or OS). This was displayed on a screen, presented for 2.5 s, containing three separate banknote-like images, with each banknote containing one of the outcome stimuli and the number of points (Fig. 1A). The position of the two gamble outcomes was randomly counter-balanced. Following a 1.5 s ISI, participants were then presented with one of the four probability stimuli, and were required to either accept or reject the gamble. Rejecting the gamble would lead to encountering the safe stimulus and collecting the number of points associated with it for that trial. Conversely accepting the gamble would lead to encountering either O1 or O2, and collecting the number of points associated with that outcome for that trial. The probability stimulus remained on the screen until the participant made a response, up to a maximum of 6 s. Then, following a 1.5 s ISI participants observed a banknote corresponding to the outcome they received, along with the number of points they collected. To encourage participants to decide at the time of probability stimulus onset, on 10% of trials, participants were not presented with a probability stimulus, and were instead required to report the reward paired with one of the outcome stimuli.

**Questionnaires.** Participants completed the following questionnaire: The Barratt Impulsivity Scale, The State-Trait Anxiety Inventory (STAI), the Penn State Worry Questionnaire (PSWQ), and the MASQ anhedonia scale. Prior to administering the task, we expected that we would identify differences in how participants treated loss and gain blocks of the task, and that this difference would be relevant for relating to the STAI, PSWQ. However, after failing to observe relevant differential behavioral treatment of gain and loss blocks, we focused only on the BIS measure and MASQ. We hypothesized that BIS would be related negatively probability prioritization. We additionally tested whether MASQ would relate negatively to reward prioritization, however did not observe this effect to be significant. Because these were planned comparisons, we do not present correction for multiple comparisons (across multiple tests), however, we note that the strength of the effect relating BIS to neural probability prioritization would survive Bonferroni correction for the two tests performed. Note that, due to an error in recording data, we failed to collect questionnaire data for one participant. Thus, Neural-Questionnaire analysis was examined for 18 participants.

## Computational models of choice data

All behavioral analysis was implemented using the Julia (version 1.5) programming language[54]. In order to gain an algorithmic description of participants decision making we fit a number of computational models to their choices. The following models are compared in Fig. 3.

For each model, we describe how it determines the probability of accepting an offer based on trial information along with model-specific free parameters.

**Expected Value Model.** The expected value model decides based on the difference in expected value for accepting and rejecting the gamble,

$$P_{accept} = logit^{-1}(\beta[P_{O1}R_{O1} + P_{O2}R_{O2} - R_{OS}]) \tag{1}$$

Here, $P_{O1}$ and $P_{O2}$ are the respective probabilities of O1 and O2 being received conditioned on accepting the gamble. $R_{O1}, R_{O2}$

and $R_{OS}$ are the number of points paired with O1, O2 and OS for that trial. $logit^{-1}$ is the standard sigmoid logistic sigmoid function. $\beta$ is a free parameter, th inverse temperature, and controls decision noise.

**Additive heuristic model.** The additive heuristic model, based on additive integration models[9,10], yet adapted for features of this task, simply does a linear integration of two features: one related to the probability of reaching the better outcome, and one related to the difference in reward between the trigger outcome and the safe outcome:

$$P_{accept} = logit^{-1}(\beta_0 + \beta_{prob}\left[P_{O_{better}} - P_{O_{worse}}\right] + \beta_{rew}[\frac{R^*_{O_{trig}}}{2} - R^*_{O_{Safe}}]) \quad (2)$$

$\beta_0 = \beta_{gain,}$ on gain trials and $\beta_0 = \beta_{loss}$ on loss trials and controls baseline tendencies to accept or reject gambles independently of trial information. $\beta_{gain}$, $\beta_{loss}$, $\beta_{prob}$, and $\beta_{rew}$ are free parameters. $P_{O_{better}}$ and $P_{O_{worse}}$ are the respective probabilities of reaching the better and worse gamble outcomes (e.g. $P_{O_{better}} = P_{O1}$ when O1 has more points). $R^*_{O_{trig}}$ is the reward of the trigger outcome (the gamble outcome – O1 or O2 – with higher absolute value), baseline corrected such that the common noise added to each item is subtracted (Fig. 1C). $R^*_{O_{safe}}$ is the reward of the safe outcome, baseline corrected such that the common noise added to each item is subtracted.

The following models are compared additionally in Supplementary Fig. 3:

**Prospect theory.** The prospect theory model (Kahneman & Tversky, 1979) allows expectations to be taken using a Probability Weighting function, $w$, and subjective utility function, $v$,

$$P_{accept} = logit^{-1}(\beta[w(P_{O1})v(R_{O1}) + w(P_{O2})v(R_{O2}) - v(R_{OS})]) \quad (3)$$

We used standard utility functions, $v(x) = x^{\alpha_{gain}}$ when $x \geq 0, v(x) = -(x^{\alpha}_{loss})$ when $x < 0$, and. We use the log odds linear Probability Weighting function, $w(p) = \frac{\delta p^{\gamma}}{\delta p^{\gamma} + (1-p)^{\gamma}}$. $\beta$, $\alpha_{gain}$, $\alpha_{loss}$, $\delta$, and $\gamma$ are free parameters.

**Sampling models.** We additionally fit two sampling models. According to our sampling models, the participant uses importance sampling to estimate the difference in utility between accepting and rejecting the gamble outcome. Both models assume participants first select a number of samples to take, $S$, which we assume is drawn from an ordered probit distribution, $OrderedProbit(S \mid n, c)$. $n$ sets the center of the distribution and is a free parameter. The scale parameter, $c$ is set to 2. Following this, the participant draws $S$ samples where each sample corresponds to either $O_1$ or $O_2$, from the distribution $q(O_i)$, which is defined below. Given $S$ samples, the participant computes an estimate of the value difference between the gamble option and safe option:

$$\hat{E} = \frac{1}{\sum_{j=1}^{S} w_j} \sum_{i=1}^{S} w_i[v(R_{o_i}) - v(R_{OS})] \quad (4)$$

$w_i$ reflects the importance weights, $w_i = \frac{P_{O_i}}{q(O_i)}$. $v$ is defined the same as it is for the prospect theory models, with two free parameters, $\alpha_{gain}$, and $\alpha_{loss}$. $R_{o_i}$ is the number of points paired with the outcome that was drawn on sample $i$.

The participant's probability of accepting is then 1 if $\hat{E} > 0$, 0 if $\overline{E} < 0$ and .5 if $\overline{E} = 0$. We define $\hat{E}$ as a function of the number of samples

taken $S$, and the number of samples drawn as $O_1$, $n_{O_1}$,

$$\hat{E}\left(n_{O_1}, S\right) = \frac{1}{n_{O_1}w_1 + \left(1 - n_{O_1}\right)w_2}[n_{O_1}w_{O_1}\left[v(R_{O_1}) - v(R_{O_{safe}})\right] \\ + [1 - n_{O_1}]w_{O_2}\left[v(R_{O_2}) - v(R_{O_{safe}})\right]] \quad (5)$$

Then the probability of acceptance then marginalizes over the number of samples taken, $S$, as well as the number of samples drawn as O1 $n_{O_1}$:

$$P_{accept} = \sum_{S=1}^{S} OrderedProbit(S|n,c)\sum_{n_{O_1}=0}^{n_{O_1}=S} \\ Binomial(n_{O_1}, s, q(O_1))P(accept|\hat{E}(n_{O_1}, S)) \quad (6)$$

where we took the maximum number of samples, $S$, to be 7. Here, we assume the number of samples taken, S, is selected from an Ordered Probit distribution, with scale parameter, c = 2, and center parameter, n, a free parameter.

We considered two sampling models, which differ with regards to the sampling distribution $q(O_i)$. For probability sampling[55], $q(O_i) \propto P_{O_i}$. For utility weighted sampling[50], $q(O_i) \propto P_{O_i}|v(R_{O_i}) - v(R_{Safe})|$. Both models have a 3 free parameters: $n$, $\alpha_{gain}$, and $\alpha_{loss}$.

**Model fitting procedure.** For each participant, we estimated the free parameters of each model by maximizing the likelihood of choices, jointly with group-level distributions over the entire population using an Expectation Maximization (EM) procedure[56]. Models were compared by computing the integrated Bayesian information criterion over the entire group of participants for each model. In order to compare model predictions to data points, we computed for each trial, for each participant, the probability of acceptance under that participant's best fitting parameters.

## MEG acquisition

MEG data was acquired on a CTF 275-channel axial gradiometer system (CTF Omega, VSM MedTech) sampling at 1200 Hz. No online filters were applied during collection. The task was divided into multiple MEG sessions, with each session lasting less then 10 minutes. Participants were asked to remain still during the scanning session. Participants were able to take a rest between sessions, however they were required to remain in place in the scanner and encouraged not to move. At the start of each scanning session participant's head positions were registered.

## MEG analysis

All MEG analyses were completed using custom Matlab (version 2019a) scripts.

**Preprocessing.** Preprocessing was performed using OSL (OHBA Analysis Group, OHBA, Oxford, UK). Preprocessing steps included high-pass filtering, at 0.5 Hz, followed by down sampling to 100 Hz. After identification and removal of excessively noisy sensors (using standard artifact rejection in OSL with default parameters – mean 8 +/– 6.01 sensors per participant), independent component analysis (ICA) was applied to denoise the data. We applied fastica, part of the AFRICA ICA procedure within the OSL software package. ICA was run with default parameters, which sets the maximum number of components that can be removed due to kurtosis to 10, and flagged components were removed automatically. Components were rejected where the kurtosis exceeded a threshold of .5. Our approach to handle eye movements is to additionally remove components that are correlated with recorded EOG channel eye movement data. The mean and standard deviation number of components removed per run is reported in Supplementary Table 3.

A set of example components removed for kurtosis is displayed in Supplementary Fig. 14. Note that to maximize power in estimating neural markers of preferential reactivation, no trials were removed due to preprocessing. We verified that this does not harm decoding accuracy (Supplementary Fig. 16). Additionally, our results are robust to altering this decision and hold under trial removal (Supplementary Fig. 17).

Data from the functional localizer task was epoched between 0 and 500 milliseconds following stimulus onset. We trained binary classifiers on data from the functional localizer task. Our decision to train classifiers from 0 ms to 500 ms post image onset was motivated by three factors. First, our analysis of classifier cross-validation accuracy revealed that we only had significant decoding (testing on the same time-point as was trained on) for train time-points from 0 ms to 560 ms post image onset. Second, prior evidence has shown that relevant reactivation events occur for classifiers trained on a time-point less than 500 ms post image onset[32]. Third, using an a priori hypothesis for which representations would be reactive allowed us to increase power for our key tests of relationships between behavioral and neural reactivation events.

**Decoding outcome stimuli.** In order to decode outcome representations, while minimizing correlations between decoded reactivations, we followed an approach recommended in[34] of training models to discriminate one state against a mixture of other states and null data. Thus, for each 10 ms timepoint following stimulus onset, three binary classifiers were trained, one for each outcome stimulus to discriminate between sensor data associated with that stimulus, and sensor data associated with each of the 6 other stimuli, along with null data corresponding to the intertrial interval (equal in number to 100% of training examples). The classification pipeline consisted of scaling the data by dividing by its 95th (absolute) percentile. Following this, data from all sensors for a given timepoint was used as training examples to train a lasso logistic regression classifier (using matlab function lassoglm). Figure 3B–D were generated by doing a 7-fold cross validation, training the three classifiers on each time-point (out of 50) using 6/7 of the training data and then testing using remaining 1/7 examples on each timepoint. The regularization hyperparameter of the logistic regression selected as the parameter which maximized the mean cross validation accuracy along the diagonal of the 2-D map in Fig. 3D (matching train and test timepoints). A given test example was considered correct if its classifier had the highest activation (out of the three). This identified .002 as the best regularization parameter, which was used for further analysis (Supplementary Fig. 4).

After choosing a lasso penalty, we trained the three classifiers on all the localizer data, on each 10-ms binned timepoint, $\tau$, following outcome stimulus onset. This generated three classifiers, one for each outcome, for each of 50 timepoints, corresponding to each 10-ms bin between 10 ms and 500 ms following outcome stimulus presentation in the localizer task. Given the task response times in addition to prior hypothesis about when relevant reactivation events occur (Ref. 32; Supplementary Fig. 5), we epoched the decision-making task data from 0 to 500 ms following the onset of the probability stimulus in each trial. We then applied each outcome classifier, for each training timepoint, $\tau$, to each task timepoint, $\tau\prime$, following probability stimulus onset. Note that, for a given trial, we only analyzed task-time points that occurred prior to a response being made (dropping time-points that occurred after this). We use $RP_{O_x}^{p,t,\tau,\tau\prime}$ to represent the reactivation probability output by the classifier, trained to activate for stimulus $O_X$ (either O1, O2, or OS) at timepoint $\tau$ ms following its presentation, for participant $s$, on trial $t$, at timepoint $\tau\prime$ following presentation of the probability stimulus.

**Relating behavioral weighting of information to preferential representation of stimuli.** To examine the question of how prioritization of

reactivated outcomes relates to behavioral evidence for reliance on probability versus reward information, we used a two-stage analysis. In the first stage, we fit, separately, for each participant, $s$, train timepoint $\tau$, and test timepoint $\tau\prime$, a linear model to predict the difference in reactivation probabilities between the two gamble outcomes, $\Delta_{RP_O}^{s,t,\tau,\tau\prime} = RP_{O_1}^{s,t,\tau,\tau\prime} - RP_{O_2}^{s,t,\tau,\tau\prime}$.

For each participant, s, train timepoint, and test timepoint, we predict this difference as a function of the participant and trial specific difference in probability, $P_{O_1}^{s,t} - P_{O_2}^{s,t}$, as well as absolute rewards, $|R_{O_1}^{s,t}| - |R_{O_2}^{s,t}|$ between the two gamble outcomes:

$$\Delta_{RP_O}^{s,t,\tau,\tau\prime} \sim \beta_0 + \beta_{prob(neural)}^{s,\tau,\tau\prime}\left[P_{O_1}^{s,t} - P_{O_2}^{s,t}\right] + \beta_{rew(neural)}^{s,\tau,\tau\prime}[|R_{O_1}^{s,t}| - |R_{O_2}^{s,t}|] \quad (7)$$

This provides an estimate of $\beta_{prob(neural)}^{s,\tau,\tau\prime}$, and $\beta_{rew(neural)}^{s,\tau,\tau\prime}$, for each participant, s, train timepoint, $\tau$, and test timepoint, $\tau\prime$. $\beta_{prob(neural)}^{s,\tau,\tau\prime}$ measures the extent to which, a tendency to reactivate O1 over O2 is driven by the probability of O1 relative to O2 (and vice-versa). Conversely, $\beta_{rew(neural)}^{s,\tau,\tau\prime}$ measures the extent to which a tendency to reactivate O1 over O2 is driven by the relative absolute reward of O1 compared to O2.

We next sought to determine whether these differences in reactivation tendencies related to behavioral reliance on reward versus probability information in choice. To examine this, in a second level, we related $\beta_{prob(neural)}^{s,\tau,\tau\prime}$ and $\beta_{rew(neural)}^{s,\tau,\tau\prime}$ to fitted parameters from the Additive Heuristic model, $\beta_{prob}$ and $\beta_{reward}$, which we now refer to as $\beta_{prob(choice)}^{s}$ and $\beta_{rew(choice)}^{s}$. We predicted that behavioral reliance on probability information, indexed by $\beta_{prob(choice)}^{s}$ would be related to preferential reactivation of more probable gamble outcomes, as indexed by $\beta_{prob(neural)}^{s}$, and that behavioral reliance on reward information, indexed by $\beta_{rew(choice)}^{s}$ would be related to preferential reactivation of outcomes with higher absolute reward, as indexed by $\beta_{rew(neural)}^{s}$.

We thus performed two between participant regressions: one relating $\beta_{prob(choice)}^{s}$ to $\beta_{prob(choice)}^{s,\tau,\tau\prime}$ (Fig. 4) and one relating $\beta_{rew(choice)}^{s}$ to $\beta_{rew(neural)}^{s,\tau,\tau\prime}$ (Fig. 5). In order to mitigate the impact of potential outliers, following previous work (Eldar et al., 2018), all between-participant behavioral-neural regressions and associated t-statistics were computed using robust linear regression, (Matlab function robustfit, with default settings). Note that this approach has been shown to both increase power and reduce false positive rates in the presence of outliers[57]. Additionally note that significance (p-values) of computed t-statistics were computed by non-parametric permutation test, thus additionally ensuring appropriate false positive rates. Specifically, these permutation tests only assume exchangeability of participants under the null distribution, which is appropriate given that participant's behavioral and neural measurements are independent of one another.

Specifically, each between participant regression was applied separately for each train timepoint, $\tau$ and test timepoint, $\tau\prime$, thus providing a 2-d map ($\tau$ by $\tau\prime$) of t-statistics for each regression. Following Ref. 32, this map was then smoothed with a Gaussian kernel ($\sigma = 1.5$ time bins). Significance for each between participant regression was computed over the peak (max) t-statistic of this smoothed map by non-parametric permutation test[32]. For this, the 2-d map was re-computed 5000 times, each time shuffling which participant was assigned to which behavioral parameter (e.g. assigning the behavioral parameter for participant 11, $\beta_{11(prob(behavior))}^{s}$, to participant 15) according to a random permutation. A null distribution over max-t-statistics was created by taking the peak of each of the 5000 t-statistic maps (over $\tau$ and $\tau\prime$). Family wise error corrected p-values ($P_{FWE}$) were computed as the proportion of permutations greater than the peak of the true observed map. We note that because all of our analysis are based on clear directional predictions, to test for these and all relationships, we employ one-tailed tests. However, we note that most reported results would still be significant under two-tailed tests.

Because the Choice Reward Weight in the additive heuristic model requires comparison of the gamble outcome with higher reward absolute value to the safe outcome, we also predicted that behavioral reward consideration would be related to reactivation of the safe outcome (Supplementary Fig. 6). As a measure of safe outcome reactivation, we computed, for each participant, $s$, train timepoint, $\tau$, and test timepoint, $\tau'$, the mean reactivation probability across trials, $RP_{O_S}^{s,\tau,\tau'}$. We then related this to $\beta_{rew(choice)}^s$ and computed significance equivalently as was done for the above between participant regressions.

In order to relate behavioral reactivation to a tendency to reinstate outcomes based on their probability (Fig. 7), we repeated the previous between participant regression involving $\beta_{prob(neural)}^{s,\tau,\tau'}$, however replacing the $\beta_{prob(choice)}^s$ with the BIS score of participant $s$. Significance of this regression was computed equivalently to the above, except here $P_{FWE}$ was computed as proportion of permutations less than the observed minimum (since a negative effect was predicted).

### Perceptual detection task procedures
This online study was approved by UCL ethics (ID: 16639/001).

**Participants.** We recruited 100 participants (mean (std) age: 27.6 (8.1), 35 female) on Prolific to perform the task online in their browser. Participants provided informed consent prior to starting the study. Data from 3 participants was lost due to errors in recording. Using an equivalent exclusion criterion as used in the MEG study, an additional 5 participants were excluded due to selection of the same action on more than 80% of trials. Finally, an additional 4 participants were removed due to failure to make responses to perceptual detection trials, leaving 88 participants for analysis. For completing the study, which took approximately 65 minutes, participants were paid 9.34 GBP, with a performance dependent bonus between 0 and 3 GBP.

**Task.** After completing instructions and passing a quiz on their contents, participants completed the BIS questionnaire followed by a probability learning task which was identical to that used in the MEG task, however only had three rather than four blocks. They then completed the risky decision-making task, consisting of 288 trials. Two thirds of trials were identical to the decision trials in the MEG task, however, were run slightly faster: with inter-stimulus intervals of 1 second and inter-trial intervals also of 1 second. Additionally, participants were only allowed to make a choice following observing the probability stimulus for 1 second.

On one third of trials, instead of being allowed to make a choice, the probability stimulus disappeared, and participants were shown the three outcome stimuli, one of which contained an arrow stimulus placed over it (Fig. 6A). Participants were then required to press an arrow key indicating the direction of the arrow as quickly as they could.

### Perceptual detection task analysis
We sought to estimate the extent to which individual participants represented outcome stimuli based on either their probability or absolute rewards. If participants tended to represent outcome stimuli based on their probability, they would make faster responses when higher probability outcome stimuli were the probed stimulus compared to when lower probability stimuli were the probe stimulus. Conversely, if participants tended to represent outcome stimuli based on their absolute rewards, they would make faster responses when higher absolute reward outcome stimuli were the probed stimulus compared to when lower absolute-reward stimuli were the probe stimulus.

We thus sought to estimate the effect of relative outcome probability and relative outcome absolute reward on log response times to the perceptual detection probe:

$$\log(rt_{s,t}) \sim \beta_0 + \beta_{prob(RT)}^s \left[ P_{O_{probed}}^{s,t} - P_{O_{non-probed}}^{s,t} \right] + \beta_{rew(RT)}^s [|R_{O_{probed}}^{s,t}| - |R_{O_{non-probed}}^{s,t}|] \tag{8}$$

Here, $rt_{s,t}$ is the response time of participant $s$ on trial $t$. $P_{O_{probed}}^{s,t}$ and $P_{O_{non-probed}}^{s,t}$ are the respective probabilities of the probed and non-probed gamble stimuli on trial for participant s, trial $t$. Note that this regression was only applied to trials where one of the gamble stimuli was the probe. Negative values of $\beta_{prob(RT)}^s$ reflect faster responses for more probable probed stimuli, reflecting a tendency to represent outcomes based on their probability. $\left|R_{O_{probed}}^{s,t}\right|$ and $|R_{O_{non-probed}}^{s,t}|$ are the respective absolute rewards paired with the probed and non-probed gamble outcomes for participant $s$ on trial $t$. Negative values of $\beta_{rew(prime)}^s$ reflect faster responses for probed stimuli with higher absolute reward, reflecting a tendency to represent outcomes based on their absolute reward.

To determine if tendencies to represent outcomes based on probability or absolute reward were related to heuristic reliance of reward and probability information in choice, we fit the additive heuristic model to participants behavior and measured spearman rank correlations to test for a relationship between $\beta_{prob(RT)}^s$ and $\beta_{prob(choice)}^s$, and between $\beta_{rew(RT)}^s$ and $\beta_{rew(choice)}^s$. Spearman rank correlation assumes a monotonic relationship between variables, which is apparent in Fig. 6B and C.

### Reporting summary
Further information on research design is available in the Nature Portfolio Reporting Summary linked to this article.

### Data availability
Raw and preprocessed MEG data have been deposited in the Open-Neuro database[58], under accession code https://doi.org/10.18112/openneuro.ds005065.v1.0.0. Behavioral data for both tasks is deposited at zenodo[59], https://doi.org/10.5281/zenodo.10950132.

### Code availability
Analysis code on github is deposited at zenodo[59]: https://doi.org/10.5281/zenodo.10950132.

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

## Acknowledgements

We thank Matt Nour, Toby Wise, Jess McFayden, Oliver Vikbladh and Rachel Bedder for helpful conversations about analysis. Additionally, we thank Daniel Bates for assistance with data collection and Nathaniel Daw for contributing code used for part of the behavioral model fitting. We acknowledge funding from the Open Research Fund of the State Key Laboratory of Cognitive Neuroscience and Learning to Y.L., a Wellcome Trust Investigator Award (098362/Z/12/Z) to R.J.D. and a Wellcome Trust Investigator Award to QJMH (221826/Z/20/Z). This work was carried out whilst R.J.D. was in receipt of a Lundbeck 20 Visiting Professorship (R290-2018-2804) to the Danish Research Centre for Magnetic Resonance. The Max Planck UCL Centre is supported by UCL and the Max Planck Society. The Wellcome Centre for Human Neuroimaging (WCHN) is supported by core funding from the Wellcome Trust (203147/Z/16/Z). The authors acknowledge support by the NIHR UCLH BRC.

## Author contributions

E.M.R., R.M., Y.L., R.J.D. and Q.J.M.H contributed to the design of the study. E.M.R. collected the data with assistance from Y.L. E.M.R. analyzed the data with feedback from R.M., Y.L., R.J.D. and Q.J.M.H. E.M.R. wrote the manuscript with input and editing from R.M., Y.L., R.J.D. and Q.J.M.H. Q.J.M.H. supervised the study.

## Competing interests

The authors declare no competing interests.
