## [Peer Review File · Nature Communications]

Heuristics in risky decision-making relate to preferential representation of informationEditorial Note: This manuscript has been previously reviewed at another journal that is not operating a transparent peer review scheme. This document only contains reviewer comments and rebuttal letters for versions considered at *Nature Communications* .

REVIEWER COMMENTS

Reviewer #2 (Remarks to the Author):

As the authors have already responded to my previous round of reviews satisfactorily, I only will refer to my remaining point about the ICA components. The authors now have presented the ICA time courses and topographies of some sample participants. This makes the whole analysis more transparent, although I still think it is not a good idea of out-projecting so much data although it is done in some software packages. However, it would have been better to present all ICA components (also the eye and cardiac components). Further, it seems that for different subjects a different number of ICA components were out-projected (for example for the last participant only 7 components are shown). Is this an error, or was the number of omitted ICA components different for each subject? If so, what was the criteria to select 7 or 8 additional components (apart from the eye and heart ICAs)?

Reviewer #3 (Remarks to the Author):

The authors adequately addressed all of my points.

Typo in line 1031: remove instead of move

Reviewer #4 (Remarks to the Author):

I was asked to step in for Reviewer 1. My sense is that Reviewer 1's comments have been largely addressed by de-emphasizing the model comparison aspects of the article. The authors argue that the article is not about testing different models of risky choice. Instead, it is about detecting whether participants are focused on reward differences or probability differences. I think that the authors also addressed the other issues related to the design of the risky choice task. The counterbalancing issue remains but I think it is an inevitable feature of this experiment and the authors have qualified the relevant conclusions related to impulsivity.

I do think that the authors could reach a wider audience if they linked their work to the emerging literature on shifts in attention and latencies in multi-attribute choice. Fisher 2021 and Yang & Krajbich 2023 have highlighted how people shift attention between attributes during the choice process. Sullivan & Huettel 2021 and Chen, Zhu, Shen, Krajbich, Hare 2023 have highlighted how certain attributes enter the choice process earlier than others. Those studies have used eye tracking, mouse tracking, and reaction times to establish how people separately consider different attributes. The current study builds on that work by showing how MEG can be used to establish how people separately consider probability or reward differences.

One major concern is with the sample size. In the main text the authors state that they anticipated needing $N=30$. Based on my understanding, $N=30$ is the minimum needed to be confident in across-subject correlations. Unfortunately, the authors were only able to collect $N=21$ and analyze 19. I understand that COVID was a factor here and so I sympathize with the authors. However, that doesn't change the fact that this study is surely underpowered for several of its main claims.

Minor comments:

I would not equate decision theory or normative behavior with expected value maximization. Expected utility theory is the normative standard, not expected value.

"would tend to represent gamble outcomes based on the absolute value of rewards and the safe outcome for comparison." - This prediction is unclear. Is it both of them separately or combined or what?

Reviewer #2

As the authors have already responded to my previous round of reviews satisfactorily, I only will refer to my remaining point about the ICA components. The authors now have presented the ICA time courses and topographies of some sample participants. This makes the whole analysis more transparent, although I still think it is not a good idea of out-projecting so much data although it is done in some software packages. However, it would have been better to present all ICA components (also the eye and cardiac components). Further, it seems that for different subjects a different number of ICA components were out-projected (for example for the last participant only 7 components are shown). Is this an error, or was the number of omitted ICA components different for each subject? If so, what was the criteria to select 7 or 8 additional components (apart from the eye and heart ICAs)?

Thank you for the suggestion for how we can make our analysis more transparent. We have edited Supplementary Figure 14 to now include all components for these participants:

Subj. 5, Session 2

Subj. 11, Session 6

Subj. 19, Session 12

Supplementary Figure 14. Representative ICA components removed for three participants.

The number of ICA components removed for kurtosis was allowed to vary between participants. Following the default settings of OSL, components were rejected where the kurtosis exceeded a threshold of .5. This occurred different number of times for different participants and thus a different number of components were rejected.

This is now clarified in the methods section:

Components were rejected where the kurtosis exceeded a threshold of .5.

Reviewer #3

The authors adequately addressed all of my points.

Typo in line 1031: remove instead of move

Thank you. This typo has been corrected.

Reviewer #4

I was asked to step in for Reviewer 1. My sense is that Reviewer 1's comments have been largely addressed by de-emphasizing the model comparison aspects of the article. The authors argue that the article is not about testing different models of risky choice. Instead, it is about detecting whether participants are focused on reward differences or probability differences. I think that the authors also addressed the other issues related to the design of the risky choice task. The counterbalancing issue remains but I think it is an inevitable feature of this experiment and the authors have qualified the relevant conclusions related to impulsivity.

Thank you for this assessment.

I do think that the authors could reach a wider audience if they linked their work to the emerging literature on shifts in attention and latencies in multi-attribute choice. Fisher 2021 and Yang & Krajbich 2023 have highlighted how people shift attention between attributes during the choice process. Sullivan & Huettel 2021 and Chen, Zhu, Shen, Krajbich, Hare 2023 have highlighted how certain attributes enter the choice process earlier than others. Those studies have used eye tracking, mouse tracking, and reaction times to establish how people separately consider different attributes. The current study builds on that work by showing how MEG can be used to establish how people separately consider probability or reward differences.

Thank you for pointing us toward this work. We agree that it is highly relevant to our study. We have now referred to this work in the introduction:

Recent research has pointed to selective consideration of different types of task information as a source of multiple forms of bias in decision making. This work has analyzed eye-tracking, mouse-tracking, and response-times to reveal that selective consideration of either some choice options, attributes, or other information can explain biases in value-based consumer choice (Krajbich et al., 2010; Krajbich & Rangel, 2011), multi-attribute choice (Sullivan & Huettel, 2021), social choice (Chen et al., 2023), intertemporal choice (Fisher, 2021), and risky decision-making (Pachur et al., 2018; Zilker & Pachur, 2022, 2023). Here, we expand on this work by first providing two new forms of evidence--based on neural decoding and behavioral priming---for a relationship between selective consideration of information and heuristic strategy in decision-making under risk. Additionally, we show an example where such selective consideration is applied to potential outcomes of a choice – thus linking this body of work with work on model-based simulation in planning that has looked at what outcomes of a choice individuals tend to consider when they decide (Callaway et al., 2022; Doll et al., 2015; Mattar & Daw, 2018; Wise et al., 2021).

We again reference this work in the discussion:

These results add to recent research demonstrating that heuristics and biases in choice can be explained for by individual differences in the types of information considered during the decision process (Chen et al., 2023; Fishera, 2021; Krajbich et al., 2010; Krajbich & Rangel, 2011; Sullivan & Huettel, 2021; Zilker & Pachur, 2022, 2023) and demonstrate a link between these processes and previous research demonstrating the role of outcome reactivation in choice evaluation (Bornstein & Daw, 2013; Castegnetti et al., 2020; Doll et al., 2015; Russek et al., 2021; Wimmer & Büchel, 2019; Wise et al., 2021).

One major concern is with the sample size. In the main text the authors state that they anticipated needing $N=30$. Based on my understanding, $N=30$ is the minimum needed to be confident in across-subject correlations. Unfortunately, the authors were only able to collect $N=21$ and analyze 19. I understand that COVID was a factor here and so I sympathize with the authors. However, that doesn't change the fact that this study is surely underpowered for several of its main claims.

Thank you for pointing out this concern. While we think that we to some extent make up for low sample size with high within-participant power (288 decision trials per participant) and the addition of a conceptually related behavioral study, we are fine with acknowledging the relatively low number of participants for the MEG study. This is now pointed out in the methods section on participants:

While this study has relatively high within-participant power (with 288 decision trials per participant), further work on this topic with higher between-participant power would be valuable.

Minor comments:

I would not equate decision theory or normative behavior with expected value maximization. Expected utility theory is the normative standard, not expected value.

Thank you for pointing out this oversight. We have replaced reference to 'expected value' in the introduction discussion of normative choice, with expected utility:

When faced with a choice among actions that can lead to multiple outcomes, decision theory postulates that individuals should compute choice values by taking the expectation over the utility of outcomes, each weighted by their probability.

We also amend below in the introduction:

Normative choice in such settings requires evaluating the gamble by summing the utility of each uncertain outcome, weighted by its probability, and comparing this expected utility to the utility of a known safe option (Bernoulli, 1954; Edwards, 1954; Savage, 1972).

"would tend to represent gamble outcomes based on the absolute value of rewards and the safe outcome for comparison." - This prediction is unclear. Is it both of them separately or combined or what?

Thank you for pointing out this lack of clarity. The prediction is that computing a difference between the reward of the item with higher absolute value and the reward of the safe item (as

postulated for reward sensitivity) requires activating the representation of the gamble outcome with higher absolute reward and also activating the safe outcome. We test for activation of the outcome of the highest absolute reward in a continuous, graded manner, by testing for an effect of a gamble outcome's relative absolute reward on its reactivation. We test whether this covaries with behavioral evidence for use of reward information. We also test for activation of the safe outcome and whether this also covaries with behavioral evidence for use of reward information.

We have amended the following paragraph of the results section to make this point more clear:

We used decoding of MEG data during choice deliberation to test whether individual variation in choice behavior was driven by differences in in which outcomes individuals tended to consider. We conjectured that individuals whose behavior reflected a greater reliance on probability information (indexed by higher Behavioral Probability Weight) would also tend to neurally represent gamble outcomes with higher probability. By contrast, individuals whose behavior reflected greater reliance of reward information (indexed by higher Behavioral Reward Weight) would tend to represent the gamble outcome with higher absolute reward and the safe outcome as this would enable them to make a comparison between the rewards of these items. Note that we will test for activation of the gamble outcome with higher probability and the gamble outcome with relative higher absolute reward in a continuous manner, looking for respective effects of probability and also absolute rewards on a tendency to reactivate those outcomes.

REVIEWERS' COMMENTS

Reviewer #2 (Remarks to the Author):

The authors had addressed all my last concerns adequately.

Reviewer #2 (Remarks on code availability):

I think the code is fine as I can judge. The files are well ordered and can be identified easily and I think it should not be a problem to run the code.

Reviewer #4 (Remarks to the Author):

The authors have addressed my concerns. However, I do think the power issue is severe enough that they should acknowledge it in the Discussion and not just buried in the Methods section. More than just saying that future work would be valuable, I think the authors should say something about how the between-participant results should be interpreted with caution.

Reviewer 4:

The authors have addressed my concerns. However, I do think the power issue is severe enough that they should acknowledge it in the Discussion and not just buried in the Methods section. More than just saying that future work would be valuable, I think the authors should say something about how the between-participant results should be interpreted with caution.

Thank you for highlighting this concern. In response, we have added the following paragraph to the discussion:

A final limitation of the MEG study is that, due to interference of the Covid-19 pandemic, the sample size (N=19) is relatively low. Although this number of participants is less than intended at study inception, we note that it is within the range for similar studies in the field^{22,45,46}. Additionally, we note that recent work examining power in between participant neuroimaging studies has shown that amount of per-participant data contributes equally to power as number of participants⁴⁷. In this regard, the amount of per-participant data in this study is relatively high, with 288 decision trials per participant. Finally, we note that an additional behavioral study provides a conceptual replication of the MEG results. Regardless however, the relatively low number of participants warrants some caution in interpretation of the MEG results. Further work on this topic with higher between-participant power would be valuable.